# Dynamic Behavior Modeling of Natural-Rubber/Polybutadiene-Rubber-Based Hybrid Magnetorheological Elastomer Sandwich Composite Structures

**DOI:** 10.3390/polym15234583

**Published:** 2023-11-30

**Authors:** Ahobal N, Lakshmi Pathi Jakkamputi, Sakthivel Gnanasekaran, Mohanraj Thangamuthu, Jegadeeshwaran Rakkiyannan, Yogesh Jayant Bhalerao

**Affiliations:** 1Department of Mechanical Engineering, Dayananda Sagar College of Engineering, Bengaluru 560078, India; ahobal-me@dayanandasagar.edu; 2School of Mechanical Engineering, Vellore Institute of Technology, Chennai 600127, India; sakthivel.g@vit.ac.in; 3Centre for Automation, Vellore Institute of Technology, Chennai 600127, India; 4Department of Mechanical Engineering, Amrita School of Engineering, Amrita Vishwa Vidyapeetham, Coimbatore 641112, India; t_mohanraj@cb.amrita.edu; 5Department of Mechanical Engineering and Design, School of Engineering, University of East Anglia, Norwich Research Park, Norwich NR47 TIJ, UK; y.bhalerao@uea.ac.uk

**Keywords:** hybrid MRE, sandwich structures, smart structures, dynamic behavior modeling, vibration, polymer composites

## Abstract

This study investigates the dynamic characteristics of natural rubber (NR)/polybutadiene rubber (PBR)-based hybrid magnetorheological elastomer (MRE) sandwich composite beams through numerical simulations and finite element analysis, employing Reddy’s third-order shear deformation theory. Four distinct hybrid MRE sandwich configurations were examined. The validity of finite element simulations was confirmed by comparing them with results from magnetorheological (MR)-fluid-based composites. Further, parametric analysis explored the influence of magnetic field intensity, boundary conditions, ply orientation, and core thickness on beam vibration responses. The results reveal a notable 10.4% enhancement in natural frequencies in SC4-based beams under a 600 mT magnetic field with clamped–free boundary conditions, attributed to the increased PBR content in MR elastomer cores. However, higher magnetic field intensities result in slight frequency decrements due to filler particle agglomeration. Additionally, augmenting magnetic field intensity and magnetorheological content under clamped–free conditions improves the loss factor by from 66% to 136%, presenting promising prospects for advanced applications. This research contributes to a comprehensive understanding of dynamic behavior and performance enhancement in hybrid MRE sandwich composites, with significant implications for engineering applications. Furthermore, this investigation provides valuable insights into the intricate interplay between magnetic field effects, composite architecture, and vibration response.

## 1. Introduction

Advances in aerospace technology have led to the development of composite materials that feature properties that rival or even surpass those of traditional materials, notably fiber-reinforced polymers (FRPs). FRPs are renowned for their remarkable attributes, including a high strength-to-weight ratio, exceptional durability, stiffness, and resistance to corrosion, wear, and impact. However, these high-performance FRP composite structures face a formidable challenge—vibrations induced by dynamic loads. These vibrations often lead to resonance conditions and the risk of catastrophic failures, exacerbated by insufficient damping characteristics. In response to these critical challenges, the development of smart materials has emerged, with alternative materials aiming to enhance the performance, structural integrity, and overall comfort of composite structures. Among these smart materials, MR materials have gained prominence due to their field-dependent rheological properties [1,2].

In the past decade, magneto-rheological materials have gained significant attention, surpassing electrorheological (ER) materials. This shift in interest is attributed to their superior yield strength, resilience to temperature variations, and tolerance to contaminants compared to ER fluids, making them ideal for controlling structural vibrations. Significant research endeavors have focused on evaluating both the properties and practical applications of MR fluids in the domain of structural vibration control [3,4]. However, the use of MR fluids is constrained by issues such as the accumulation of iron particles in the absence of a magnetic field and their relatively high production costs. While MR gels and grease offer an exceptional performance, they are susceptible to issues like sedimentation, deposition, environmental pollution, and sealing problems [5,6]. In contrast, MR elastomers (MREs), featuring rubber as their matrix material, excel in surmounting these challenges. They exhibit swift and reversible transformations, tunable stiffness, and advantageous viscoelastic properties in response to magnetic field application [7]. MREs provide several advantages, including lower manufacturing costs and the absence of iron particle accumulation, making them an appealing choice for various engineering applications.

In the development of magnetorheological elastomers (MREs), their composition plays a pivotal role in shaping their characteristics. Chen et al. [8] emphasized the significant impact of both the applied magnetic field and the iron particle content on MRE damping properties, offering insights into the intricate interplay between magnetic field intensity and material composition. The essential structure of MREs, characterized by a matrix of rubber intricately mixed with dispersed magnetic particles, has been elucidated by various researchers [9,10], highlighting the critical role of this composite structure in shaping MRE behavior. Several variables come into play when molding the properties of MREs, including the choice of magnetic filler material, the matrix type, and the compatibility of the magnetic filler with the matrix. Typically, iron particles of diverse shapes and sizes, exhibiting ferromagnetic properties with high magnetic saturation and soft magnetic attributes, emerge as the preferred choice for MREs [8,11]. Furthermore, the exceptional magnetorheological performance of MREs can be attributed to the synergy between magnetic fillers and the matrix material [12]. Chen et al. has demonstrated that natural-rubber-based MREs surpass silicone-rubber-based counterparts in terms of various properties, including their tear strength, tensile strength, resilience factor, and hardness [13]. Investigating isotropic synthetic-rubber-based MREs, as shown by Gong et al., shows an impressive 26% enhancement in the magnetorheological (MR) effect with the incorporation of a 0.6 volume fraction of carbonyl iron (CI) particles. This enhancement undoubtedly signifies a marked improvement in the material’s rheological behavior. However, it does introduce a trade-off, as it entails a reduction in the material’s elastic properties [14]. This reduction in elasticity could be of concern, especially in structural applications where the ability to isolate vibrations is of paramount importance. MREs with diminished elastic properties may struggle to obtain the necessary stiffness and structural integrity required to safeguard the overall system’s performance and reliability.

To overcome the limitations posed by the reduced elastic properties in magnetorheological elastomers (MREs), extensive research has focused on incorporating various additives, including plasticizers, silane coupling agents, and nano-sized particles like carbon black, carbon nanotubes, and graphene [15,16,17,18,19]. These additives serve to enhance the mechanical properties of MREs by reinforcing the interfacial interactions between fillers and the elastomeric matrix; however, they are not exempt from significant challenges. One of these challenges lies in the propensity of these fillers, particularly when in nano-sized forms, to agglomerate within the elastomeric matrix [20]. This phenomenon not only complicates the manufacturing process but also increases costs, particularly in the case of nanomaterials [21]. Ensuring compatibility between the chosen fillers and the matrix material, as well as addressing processing intricacies and potential health and environmental concerns, emerges as crucial considerations in MRE development. Achieving the delicate equilibrium between attaining desired properties and managing these inherent limitations constitutes a challenging task. Consequently, researchers are compelled to continually explore novel materials and innovative techniques, with the aim of optimizing MRE composites for a diverse array of applications.

In recent years, researchers have embarked on an exploration of the untapped potential within hybrid matrix magnetorheological elastomer (MRE) composites, aiming to conquer the persistent challenges posed by nanofillers and the relatively modest mechanical properties inherent in conventional MREs. This innovative approach has opened up a promising pathway for addressing the challenges associated with nanofillers and the relatively low mechanical properties observed in conventional MREs [22,23,24].

Researchers aimed to find a balance between the mechanical performance and magnetorheological (MR) effect by using NR and PBR by addressing a common challenge in MRE development. NR-based MREs excel in terms of their mechanical properties but often fall short in terms of the required MR effect for industrial applications. In contrast, PBR-based MREs exhibit a high MR effect but suffer from inferior mechanical properties. The synergy between NR’s excellent synthetic mechanical performance and PBR’s desirable characteristics, including high elasticity, low heat buildup, cold resistance, and flex fatigue resistance, allows for the development of hybrid matrices that capitalize on the strengths of both materials. This harmonious blend results in improved compatibility, as NR and PBR share active cross-linking spots and possess similar vulcanization mechanisms and curing rates [25]. Song et al.’s research reveals that increasing polybutadiene rubber (PBR) content from 10% to 50% results in a minor decrease in the zero-field modulus, accompanied by a substantial enhancement in the magneto-rheological (MR) effect, ranging from 31.25% to 44.19% [25]. This work underscores the trade-off between these critical material properties in hybrid MREs, with implications for future applications. Several works on matrix materials incorporating blends of NR/styrene–butadiene rubber (SBR) and NR/nitrile butadiene rubber (NBR) reveals improved mechanical properties when compared to a matrix composed of only NR or NBR [26]. Pal et al. showed that blends of urethane rubber (UR)/NR and PBR/NR exhibited a 35–40% enhancement in mechanical properties [27]. The study by Ge et al. revealed that the incorporation of rosin glycerin ester into natural rubber/rosin glycerin hybrid-matrix-based MREs resulted in an increase in the zero-field modulus (G_0_) at a 9% concentration, but this effect diminished at higher concentrations. Furthermore, an increase in carbon iron (CI) content led to a substantial 575% improvement in G_0_ at 80 wt%. Additionally, the application of a magnetic field intensified interparticle forces within CI-based MREs, highlighting the potential for tailoring the mechanical properties of these materials for diverse engineering applications [28].

Previous research into the hybrid matrix MREs has made limited progress in comprehending their mechanical properties, rheological properties and dynamic behavior. There is a clear absence of both mathematical approaches and experimental investigations analyzing the dynamic behavior of hybrid MRE composites. In the present study, the dynamic characteristics of NR/PBR-based hybrid magnetorheological (MR) elastomer sandwich composite beams are investigated using numerical simulations. This investigation considers various compositions of NR and PBR to develop finite element equations for the sandwich composite beam. The potential energy and kinetic energy equations for the hybrid elastomer sandwich composite beam with FRP face sheets, employing Reddy’s third-order shear deformation theory, are derived. Remarkably, there is a lack of research on the dynamic study of hybrid MRE sandwich composite structures. To address this gap, the governing differential equations of motion for the hybrid MRE sandwich composite beam are established, considering various compositions of the NR/PBR matrix. These equations are presented using a three-node line element with five degrees of freedom at each node. To validate the finite element simulations, the results are compared with the existing data on MR fluids that are available in the literature. Additionally, the study examines the dynamic characteristics of various configurations of hybrid MRE sandwich composite beams under the influence of magnetic field intensity, ply orientation, core thickness, and boundary conditions. Figure 1 outlines the various steps involved in the numerical simulation of hybrid MRE sandwich composites, representing a concerted effort to gain a deeper understanding of the characteristics of these structures.

## 2. Mathematical Modelling

Sandwich composite structures with laminated face sheets, as shown in Figure 2, are used in various high-performance engineering applications, such as aircraft wings, windmill blades, and helicopter rotor blades, due to their exceptional structural characteristics. These composite structures consist of a core composed of carbonyl iron powder (CIP), a magnetic filler material uniformly dispersed in a hybrid MR elastomer core comprising both polybutadiene rubber (PBR) and natural rubber (NR). Flanking this core are multi-layered face sheets made of glass-fiber-reinforced polymer. In the analysis, perfect bonding is assumed between all three layers of the sandwich composite beam. To comprehensively explore the dynamic properties of these composite beams, four distinct configurations of hybrid MR elastomers, denoted as SC1, SC2, SC3, and SC4, are considered. The composition of these hybrid elastomers is detailed in Table 1, where the proportions of PBR mixed with NR are determined following the approach outlined by Song et al. [25]. The geometric parameters of the sandwich composite beam include the length and width of the face sheets, represented as ‘L’ and ‘w’, respectively. Additionally, we define the thicknesses of the elastomer core, bottom face sheet, and top face sheet as ‘*h_c_*’, ‘*h_b_*’, and ‘*h_t_*’, respectively.

### 2.1. Modelling of Face Sheets

The governing differential equations of motion for the sandwich composite beam were formulated using Reddy’s third-order shear deformation theory (RTSDT). RTSDT incorporates shear deformation effects by assuming that the deformation field of the skin layers varies as a third-order function of $x_{3}$, representing the thickness coordinate of the sandwich composite beam. This theory provides a more accurate description of the displacement field for the face sheets of the sandwich composite beam, and its formulation is as follows:(1)$${u_{1}}^{j}\left(x_{1},x_{3},t\right)={u_{1}}_{0}^{j}\left(x_{1},t\right)+x_{3}\varphi \left(x_{1},t\right)+{x_{3}}^{2}\theta \left(x_{1},t\right)+{x_{3}}^{3}\emptyset \left(x_{1},t\right)\;j=t,b\;{u_{3}}^{j}\left(x_{1},x_{3},t\right)={u_{3}}_{0}^{j}={u_{3}}_{0}$$

The displacements along the $x_{1}$-axis for the first and third layers, as well as the transverse deflection along the $x_{3}$-direction, are denoted as ${u_{1}}^{j}$ and ${u_{3}}^{j}$, respectively. It is assumed that the transverse deflection for the first and third layers is equal. Furthermore, considering that the bottom and top layers of the composite beam are traction-free, the boundary conditions can be expressed as follows:(2)$$\tau_{x_{1}x_{3}}^{j}\left(x_{1},\pm \frac{h_{j}}{2},t\right)=0$$

To account for the assumed traction-free boundary condition of the beam, we can simplify the displacement parameters by eliminating the second-order terms. Following this reduction, the resulting displacement field, which contains third-order terms, is given as follows:(3)$${u_{1}}^{j}\left(x_{1},x_{3},t\right)={u_{1}}_{0}^{j}\left(x_{1},t\right)+x_{3}\varphi \left(x_{1},t\right)-\frac{4{x_{3}}^{3}}{3{h_{j}}^{2}}\zeta \;\zeta =\left(\frac{\partial {u_{3}}_{0}^{j}}{\partial x_{1}}+\varphi \right)$$

The transverse term about the *x*_1_-axis is represented as φ, while the wrapping term ζ is expressed in terms of the transverse plane rotation. The strain-displacement relationship for the face sheets of the sandwich composite beam, derived using the small strain theory, is as follows:(4)$$\varepsilon_{x_{1}}^{j}=\varepsilon_{0}^{j}+x_{3}\varepsilon_{1}^{j}+{x_{3}}^{3}\varepsilon_{3}^{j},\;\gamma_{x_{1}x_{3}}^{j}=\gamma_{0}^{j}+{x_{3}}^{2}\gamma_{2}^{j}$$
where
(5)$$\varepsilon_{0}^{j}=\frac{\partial {u_{1}}_{0}^{j}}{\partial x_{1}},\;\varepsilon_{1}^{j}=\frac{\partial \varphi }{\partial x_{1}},\;\varepsilon_{3}^{j}=-k_{1}\frac{\partial \zeta }{\partial x_{1}}\;\gamma_{0}^{j}=\zeta ,\;\gamma_{2}^{j}=-k_{2}\zeta$$
where $\;k_{1}=4/3{h_{j}}^{2}$ and ${k_{2}=3k}_{1}$. The resultant force and moments of the sandwich composite beam, which are analogous with the strain terms given in Equation (4), are represented as:(6)$$\left\{N^{j}\right\}=\left[ABD\right]\left\{\varepsilon^{j}\right\}$$
where the $\left\{N^{j}\right\}$ matrix includes bending moment, in-plane force, transverse shear and higher-order shear force resultants, and higher-order moments. The $\left[ABD\right]$ matrix contains terms of transverse shear stiffness, bending stiffness, extensional stiffness, extensional–bending stiffness, higher-order transverse shear stiffness and extensional bending, which are provided in Equation (A1) of Appendix A.

### 2.2. Modelling of MR Elastomer Core

In the modeling of magnetorheological elastomers (MREs), it is presumed that the normal stresses that developed within the MR elastomer core are negligible due to its significantly lower elastic modulus compared to the face sheets. Additionally, a no-slip condition is assumed at the interface between the face sheets and the MR elastomer core, simplifying the boundary conditions. To facilitate the modeling process, the longitudinal deformation (${u_{1}}^{c})$ and transverse shear strain ($\gamma^{c}$) of the elastomer core are derived from the kinematics of the deformed elastomer, providing essential parameters for the construction of a comprehensive MRE model.
(7)$${u_{1}}^{c}=\frac{{u_{1}}^{t}-{u_{1}}^{b}}{2}+\left(\frac{h_{b}-h_{t}}{4}\right)\varphi +\left(\frac{h_{b}-h_{t}}{12}\right)\zeta \;\gamma^{c}=\frac{{u_{1}}^{t}-{u_{1}}^{b}}{h_{c}}-\left(\frac{h_{t}+h_{b}}{2h_{c}}\right)\varphi +\left(\frac{h_{t}+h_{b}}{6h_{c}}\right)\zeta +\frac{\partial {u_{3}}_{0}}{\partial x_{1}}$$

The resultant shear force associated with MR elastomer core is expressed as
(8)$$Q^{c}=G_{xz}^{\prime }h_{c}\gamma^{c}$$
where $G_{xz}^{\prime }$ is a complex shear modulus of the constraining layer, which is expressed as
(9)$$G_{xz}^{\prime }=G^{*}(1+iɳ)$$

### 2.3. Kinetic and Strain Energy Formulations

The strain energy (δU) attributed to the virtual displacements in the NR/BR hybrid MR elastomer sandwich composite beam with face sheets can be expressed as follows:(10)$$\delta U=\int w\left(\sum_{j=t,b}\left(N^{j}{\delta \varepsilon }_{0}^{j}+M^{j}{\delta \varepsilon }_{1}^{j}-k_{1}{\delta P}^{j}\varepsilon_{3}^{i}+Q^{j}{\delta \gamma }_{0}^{j}-k_{2}R^{j}\delta \gamma_{2}^{j}\right)+R^{c}\delta \gamma^{c}\right)dx_{1}$$

Furthermore, the strain energy is reduced in relation to the deformation field, as provided in Appendix A Equation (A3).

The kinetic energy (δK) resulting from the in-plane, transverse, and shear displacements of the structure is expressed as follows:(11)$$\begin{array}{ll} \delta K= & \sum_{j=t,b}\int_{h_{b}^{j}}^{h_{t}^{j}}w\rho_{f}[(({{\dot{u}}_{1}}_{0}^{j}+x_{3}\dot{\varphi }-k_{1}\dot{\zeta })\left({\delta \dot{u_{1}}}_{0}^{j}+x_{3}\dot{\delta \varphi }-k_{1}\dot{\delta \zeta }\right)+{\dot{u_{3}}}_{0}\delta {\dot{u_{3}}}_{0})\\ & +{x_{3}}^{2}\left(\frac{\partial {\dot{u_{3}}}_{o}^{j}}{\partial x_{1}}+\dot{\varphi }+{k_{2}x_{3}}^{2}\dot{\zeta }\right)\left(\frac{\partial \delta {\dot{u_{3}}}_{o}^{j}}{\partial x_{1}}+\delta \dot{\varphi }+{k_{2}x_{3}}^{2}\delta \dot{\zeta }\right)]dx_{1}dx_{3}\\ & +\int_{-\frac{h_{c}}{2}}^{\frac{h_{c}}{2}}{w\rho }_{c}\left({\dot{u}}_{c}\delta {\dot{u}}_{c}+{\dot{u_{3}}}_{0}\delta {\dot{u_{3}}}_{0}+{x_{3}}^{2}\gamma^{c}{\delta \gamma }^{c}\right)\;dx_{1}dx_{3} \end{array}$$

The kinetic energy equation, in reduced form, is provided in Appendix A Equation (A4).

The expression for the virtual work carried out (δV) due to the distributed transverse load q(x,t) at time *t* is as follows:(12)$$\delta V=\int q\delta {u_{3}}_{0}dx_{1}$$

Let $\delta w_{0}$ represent the virtual transverse deflection of the sandwich composite beam. When dealing with dynamic structures, it is important to establish that admissible virtual displacements are set to zero at two specific instances, denoted as $t_{1}$ and $t_{2}$, during which the precise position of the structure is known. To derive the variational functional, denoted as *I*, for the initial value problem, Hamilton’s principle is employed as follows:(13)$$\begin{array}{ll} I= & \int_{t1}^{t2}\int [-\left({\frac{\partial N}{\partial x_{1}}}^{t}-I_{0}{\ddot{u_{1}}}_{0}^{t}-I_{1}\ddot{\varphi }+k_{1}I_{3}\ddot{\xi }\right)\delta {u_{1}}_{0}^{b}-\left({\frac{\partial N}{\partial x_{1}}}^{b}-I_{0}{\ddot{u_{1}}}_{0}^{b}-I_{1}\ddot{\varphi }+k_{1}I_{3}\ddot{\zeta }\right)\delta {u_{1}}_{0}^{b}\\ & -\left(\frac{\partial M^{t}}{\partial \partial x_{1}}+\frac{\partial M^{c}}{\partial x_{1}}-Q^{t}-Q^{b}-{2I}_{2}\ddot{\varphi }-I_{1}\left({\ddot{u_{1}}}_{0}^{t}+{\ddot{u_{1}}}_{0}^{b}\right)+{2k}_{1}I_{4}\ddot{\zeta }\right)\delta \varphi \\ & -\left(\frac{\partial Q^{t}}{\partial x_{1}}+\frac{\partial Q^{b}}{\partial x_{1}}+\frac{\partial Q^{c}}{\partial x_{1}}+q-I_{0}{\ddot{u_{3}}}_{0}\right)\delta {u_{3}}_{0}\\ & -\left(-k_{1}\frac{\partial P}{\partial x_{1}}+\frac{4R}{{h_{j}}^{2}}+k_{1}I_{3}\left({\ddot{u_{1}}}_{0}^{t}+{\ddot{u_{1}}}_{0}^{b}\right)+2k_{1}I_{4}\ddot{\varphi }+{k_{1}}^{2}I_{6}\ddot{\zeta }\right)\delta \zeta ]dx_{1}\;dt \end{array}$$

The governing differential equations of motion of the sandwich composite beam are attained by setting the coefficients of virtual displacements in the domain Ω to zero.
(14)$$\delta {u_{1}}_{0}^{t}:{\frac{\partial N}{\partial x_{1}}}^{t}-I_{0}{\ddot{u_{1}}}_{0}^{t}-I_{1}\ddot{\varphi }+k_{1}I_{3}\ddot{\zeta }=0\;\delta {u_{1}}_{0}^{b}:{\frac{\partial N}{\partial x_{1}}}^{b}-I_{0}{\ddot{u_{1}}}_{0}^{b}-I_{1}\ddot{\varphi }+k_{1}I_{3}\ddot{\zeta }=0\;\delta \varphi :\frac{\partial M^{t}}{\partial x_{1}}+\frac{\partial M^{b}}{\partial x_{1}}-Q^{t}-Q^{b}-{2I}_{2}\ddot{\varphi }-I_{1}\left({\ddot{u_{1}}}_{0}^{t}+{\ddot{u_{1}}}_{0}^{b}\right)+{2k}_{1}I_{4}\ddot{\zeta }=0\;\delta {u_{3}}_{0}:\frac{\partial Q^{t}}{\partial x_{1}}+\frac{\partial Q^{c}}{\partial x_{1}}+\frac{\partial Q^{b}}{\partial x_{1}}+q-I_{0}{\ddot{u_{3}}}_{0}\;\delta \zeta :-k_{1}\left(\frac{\partial P^{t}}{\partial x_{1}}+\frac{\partial P^{b}}{\partial x_{1}}\right)+\frac{4}{{h_{j}}^{2}}\left(R^{t}+R^{b}\right)+k_{1}I_{3}\left({\ddot{u_{1}}}_{0}^{t}+{\ddot{u_{1}}}_{0}^{b}\right)+2k_{1}I_{4}\ddot{\varphi }+{k_{1}}^{2}I_{6}\ddot{\zeta }=0$$

### 2.4. Finite Element Formulations

The sandwich composite beam, which consists of multiple layers of an NR-BR hybrid magnetorheological elastomer (MRE) with face sheets, was modelled using elements featuring three nodes, each with five degrees of freedom (DOF), as illustrated in Figure 3. This beam element model encompasses various parameters at each node, including axial deformations of the top (${u_{1}}_{0}^{t}$), and bottom $\left({u_{1}}_{0}^{b}\right)$ face sheets, transverse deflection (${u_{3}}_{0})$, transverse rotation (φ), and a higher-order term (ζ). To represent the deformation field within this distinctive element of the sandwich composite beam, we utilized nodal DOF and Lagrange interpolation functions in natural coordinates as follows:(15)$$\left\{\begin{array}{c} {u_{1}}_{0}^{t}\\ {u_{1}}_{0}^{b}\\ \begin{array}{c} {u_{3}}_{0}\\ \varphi \\ \zeta \end{array} \end{array}\right\}=\left[\begin{array}{ccccc} {\hat{N}}_{k} & 0 & 0 & 0 & 0\\ 0 & {\hat{N}}_{k} & 0 & 0 & 0\\ 0 & 0 & {\hat{N}}_{k} & 0 & 0\\ 0 & 0 & 0 & {\hat{N}}_{k} & 0\\ 0 & 0 & 0 & 0 & {\hat{N}}_{k} \end{array}\right]\left\{\begin{array}{c} {u_{1}}_{0k}^{t}\\ {u_{1}}_{0k}^{b}\\ \begin{array}{c} {u_{3}}_{0k}\\ \varphi_{k}\\ \zeta_{k} \end{array} \end{array}\right\}\;k=1,2,3$$

Substituting Equation (12) into the variational principle, and expressing this in terms of the finite element equation, the governing equations of motion are written as follows:(16)$${\left[M\right]}^{e}{\left\{\ddot{d}\right\}}^{e}+\left[K^{*}\right]{\left\{d\right\}}^{e}={\left\{f\right\}}^{e}$$
where ${\left\{d\right\}}^{e}$, ${\left[M\right]}^{e}$, ${\left\{f\right\}}^{e}$, and $\left[K^{*}\right]$ denote the element deformation vector, element mass matrix, element force vector and element stiffness matrix, respectively. After the assembly of the element mass matrix and stiffness matrix, the governing equations of motion for composite beam is provided as follows:(17)$$\left[M\right]\left\{\ddot{q}\right\}+\left[K^{*}\right]\left\{q\right\}=\left\{f\right\}$$

$\left\{d\right\}$, $\left[M\right]$, $\left\{f\right\}$, and $\left[K^{*}\right]$ denote the global deformation vector, mass matrix, force vector, and global complex stiffness matrix respectively. The force vector for free vibration is considered to be null, thereby reducing the Equation (15) as follows:(18)$$\left[M\right]\left\{q\right\}+\left[K^{*}\right]\left\{q\right\}=\left\{0\right\}$$

The solution $\left\{d\right\}$ to Equation (17) can be expressed in terms of arbitrary constant ($C_{1}$) as follows:(19)$$\left\{q\right\}=[C_{1}]\left\{e^{\lambda t}\right\}$$

Equation (19) reduces the Equation (18) into an eigen value problem as follows:(20)$$\left[\left[M\right]-\lambda \left[K^{*}\right]\right]\left\{C_{1}\right\}=0$$
where λ is the characteristic value, which is obtained as follows:(21)$$\lambda =\left[\ddots \lambda_{j}\ddots \right]$$

The physical deformation vector can only be determined using Equation (19) after formulating the deformation vector q(t) from Equation (20).

The natural frequency ($\omega_{j})$ and loss factor (${ɳ}_{j})$ at each mode are obtained using Equation (21), as follows:(22)$$\omega_{j}=\sqrt{Re(\lambda_{j})}\;{ɳ}_{j}=\frac{Im(\lambda_{j})}{Re(\lambda_{j})}$$

### 2.5. Validation of Established Finite Element Formulation

The validity of finite element simulations is verified by comparing the natural frequencies of the MR fluid sandwich beam with the elastic face layers obtained from the available literature using the developed MATLAB code. The geometrical and mechanical properties of the sandwich composite are considered to be same as those of Rajmohan et al. [29]. Elastic layer length = 300 mm; breadth b = 30 mm; thickness h_e_ = 1 mm; elastic modulus E_e_ = 68 GPa; storage modulus G_e_ = 26 GPa; density of elastic layer ρ_e_ = 2700 kg/m^3^; MR fluid core thickness h_c_ = 1 mm; density of rubber ρ_r_ = 1233 kg/m^3^; density of MR fluid ρ_cf_ = 3500 kg/m^3^; shear modulus function of MR fluid G′ = −3.3691G^2^ + 4.9975 × 10^3^G + 0.893 Mpa; G″ = 0.9 G^2^ + 0.8124 × 10^3^G + 0.1855 M Pa, where G indicates the applied magnetic field intensity in Gauss. The simulations were performed to obtain the first five natural frequencies of the three-layer MR fluid sandwich composite at 0 G and 250 G magnetic fields under simply supported boundary conditions. The simulated natural frequencies are correlated with those of Rajmohan et al. [29] and Rajmohan et al. [30], as presented in Table 2. The maximum deviation observed between the simulated frequencies with the frequencies presented by Rajmohan et al. [29] is 8%, and with that observed with the frquencies of Rajmohan et al. [30] is 4%. The models used in Rajmohan et al. [29] and Rajmohan et al. [30] assume there was no warping of transverse normal during the deformation resulting in deviations in the results. Therefore, the FE model that was developed based on the Reddy third-order shear deformation can be considered to be a better model for evaluating the dynamic characteristics of a sandwich composite beam. Further, the conclusion may be drawn that the proposed model has good agreement with Rajmohan et al. [29] and Rajmohan et al. 30] in predicting the natural frequencies of three-layer MR fluid composite beam.

In addition to using finite element simulations for predicting and comprehending the dynamic behavior of MR elastomer sandwich composite structures, the validation of the FEM approach is further substantiated by comparing the loss factor with data obtained by RajaMohan et al. [30] for MRE sandwich composite structures as shown in Figure 4. This comprehensive validation underscores the accuracy of the finite element model in capturing intricate interactions and behaviors within such composite structures. By achieving a strong alignment with prior research, the model instils confidence in its ability to predict and analyze the dynamic responses of similar MR elastomer sandwich composite structures.

## 3. Results

Finite element simulations are performed on a hybrid MRE sandwich composite beam with dimensions of 300 mm × 30 mm × 4 mm to evaluate loss factor and natural frequencies. The face sheet is assumed to be a three-layer fibre-reinforced polymer laminate with the layup sequence of [0°/90°/0°]_s_ and 0.54 mm thickness. The hybrid elastomer thickness is considered to be 3 mm. The mechanical properties of the face sheet [31] and rheological properties of the hybrid elastomer [26] considered for the simulation are presented in Table 3 and Table 4, respectively. The analysis was performed under a magnetic field intensity that varied from 0 mT to 750 mT and three different end conditions: both ends clamped (CC), one end clamped and free at another end (CF), and simply supported at both ends (SS). The damping characteristics and natural frequencies of the sandwich composite beam are highly influenced by the matrix mixture (NR/PBR) content, magnetic field intensity, Ply orientation, thickness ratio and boundary conditions. The change in loss factor has a substantial effect on the damping properties of the composite beam. Therefore, it is essential to examine the influence of all these parameters on the dynamic properties of the sandwich composite beam.

### 3.1. Influence of Applied Magnetic Field Intensity on Natural Frequency and Loss Factor of Sandwich Composite Beam

The investigation is conducted to examine the effect of applied magnetic field intensity on the natural frequencies and loss factor of a sandwich composite beam. The extracted fundamental natural frequencies of sandwich composite beams with various hybrid elastomer cores, such as SC1, SC2, SC3 and SC4, are plotted for various magnetic field and boundary conditions, as shown in Figure 5. Also, the first five natural frequencies of various sandwich composite beams for various magnetic field and boundary conditions are presented in Table 5. It can be clearly seen that, at a zero magnetic field, the fundamental natural frequency increases by 2% under CC (Figure 5a), 4% under SS (Figure 5b) and 3% under CF (Figure 5c) boundary conditions, respectively, with an increase in NR content from 0 to 90 phr. The increment in the natural frequencies of all configurations could be due to improvements in the stiffness of the hybrid MR elastomer resulting from increased strain crystallization with the increase in NR content [26]. The maximum increase in the fundamental natural frequency of SC1-, SC2-, SC3- and SC4-based composite beams was found to be 7.6%, 8.5%, 10.1% 10.4%, respectively, at 600 mT under the CF boundary condition, as shown in Figure 5c. The highest improvement in fundamental natural frequency was observed in the SC4-based composite beam. This could be due to the increased compatibility of CI particles with PBR, which restricts the motion of rubber molecules, resulting in an improved field-dependent modulus. In Figure 5a–c, with the applied magnetic field intensity, the natural frequencies at all modes tend to increase up-to 600 mT, and it can further be observed that there is a slight decrement at a higher-intensity magnetic field. The cause of this decrement at a higher magnetic strength can be ascribed to the agglomeration of magnetic filler particles and breakdown of separated filler chains [32]. Also, the results indicate that CC and CF boundary conditions have the highest and least natural frequencies, respectively, under all intensities of magnetic field. This can be attributed to the fact that the clamped ends provide a higher stiffness to the beam when compared to the free end.

The loss factor at the fundamental mode of various sandwich composite beams is plotted for various boundary conditions with the magnetic field intensity shown in Figure 6. Referring to Figure 6c, the improvement in the loss factor increases from 66% to 136% with the increase in magnetic field intensity from 0 to 600 mT when PBR content increases from 10% to 100% under the CF boundary condition. This increase in damping could be due to the increased interfacial friction between the hybrid matrix and CI particles resulting from intermolecular interactions as the PBR content increases. Further, the loss factor for SC1-, SC2-, SC3- and SC4-based composite beams was found to decrease by 54%, 50%, 44% and 43%, respectively, when the magnetic field increases from 0 mT to 600 mT for CF boundary conditions. Under the externally applied magnetic field, the alignment of CI particles restricts the movement of hybrid rubber matrix molecules, resulting in reduced energy dissipation and thereby reducing the loss factor [33]. It can be observed that SS and CC boundary conditions have the largest and lowest loss factor, respectively, under all magnetic field intensities.

### 3.2. Influence of MRE Core Thickness on Natural Frequencies and Loss Factor of Sandwich Composite Beam

The effect of the MRE core thickness on the fundamental loss factor and natural frequency of sandwich composite beams is summarized in Table 6 and Table 7. It is evident that, as the core thickness increases from 1.5 mm to 4.5 mm, the fundamental natural frequency of composite beams for SC1, SC2, SC3, and SC4 increases by approximately 16%, 15.5%, 14.8%, and 14.2%, respectively, under zero magnetic field conditions with clamped-free (CF) boundary conditions. This increase in natural frequency could be attributed to the greater stiffness of the beam resulting from the thicker core. Furthermore, Table 7 reveals that the loss factor for composite beams SC1, SC2, SC3, and SC4 increases by 75%, 78%, 80%, and 84%, respectively, at a zero magnetic field under CF boundary conditions. This significant improvement in damping properties is likely due to the increased interfacial friction between the hybrid matrix and CI particles within the thicker core. These findings underscore the potential to optimize the mechanical and damping properties of sandwich composite beams by carefully selecting and adjusting the thickness of the MRE core, offering valuable insights for engineering applications requiring vibration control and damping.

### 3.3. Influence of Ply Orientation Natural Frequency and on Loss Factor of Sandwich Composite Beam

The study focuses on the damping characteristics and natural frequency of sandwich composite beam samples with three different ply orientations: [0°/90°/0°]_s_, [90°/0°/90°]_s_, and [0°/90°/45°]_s_, and their respective effects are summarized in Table 8 and Table 9. It is noteworthy that the fundamental natural frequency of all beam configurations at a zero magnetic field, under clamped-free (CF) boundary conditions, follows the order of ply orientation: [90°/0°/90°]_s_, [0°/90°/45°]_s_, and [0°/90°/0°]_s_. Specifically, the natural frequency increases by 11% for [0°/90°/0°]_s_ and 10% for [0°/90°/45°]_s_ compared to [90°/0°/90°]_s_ across all composite beam configurations. The lowest natural frequency is associated with the ply orientation [90°/0°/90°]_s_, signifying that the outermost face-sheet with a 90° orientation contributes to decreased beam stiffness. Furthermore, under zero-magnetic-field and CF boundary conditions, the loss factor for all beam configurations follows the order of ply orientation: [90°/0°/90°]_s_, [0°/90°/0°]_s_, and [0°/90°/45°]_s_. Notably, the loss factor increases by 1% for [0°/90°/0°]_s_ and 15% for [0°/90°/45°]_s_ compared to [90°/0°/90°]_s_ across all composite beam configurations. While [0°/90°/45°]_s_ exhibits a high loss factor, the natural frequency remains relatively lower due to the 45° ply introducing shear forces, which reduce beam stiffness. The increase in the loss factor could be attributed to the heightened energy dissipation facilitated by the unrestricted motion of rubber molecules, caused by the presence of carbonyl iron (CI) particles in the hybrid matrix. These results collectively demonstrate the intricate interplay between ply orientation, natural frequency, and loss factor, shedding light on the dynamic behavior of the sandwich composite beams.

### 3.4. Frequency Response of Hybrid MRE Sandwich Composite Beam

The investigation focused on the transverse vibration response of the hybrid MRE sandwich composite beam (SC1) under the constraint-free (CF) boundary condition, with varying magnetic field intensities. The responses were analyzed across a frequency range of 1–250 Hz by considering the harmonic excitation force of magnitude 5 N at the free-end corner of the beam. Various forced vibration simulations were performed, as depicted in Figure 7. In Figure 7, it is evident that there is a noticeable right shift in the natural frequency at higher frequencies as the magnetic field intensity increases. This phenomenon can be attributed to appreciation in the stiffness with the increase in magnetic field strength. Additionally, it can be observed that the amplitude of vibration decreases with higher magnetic field intensities. Figure 8 presents the vibration responses of all configurations of composite beams at 450 MT. Due to variations in the stiffness of the beams, slight fluctuations in responses at certain modes can be observed. These results imply that the application of a magnetic field has a significant impact on the transverse vibration behavior of the hybrid MRE sandwich composite beam. As the magnetic field intensity increases, the natural frequencies shift to higher values, and the amplitudes of vibration decrease. This finding suggests the potential for precise control and tuning of the dynamic response of such composite structures, which is promising for various engineering applications where vibration control and damping are crucial.

## 4. Conclusions

The paper extensively investigates the dynamic characteristics of MR hybrid sandwich composite beams with various configurations. It utilizes differential equations based on the Reddy third-order shear deformation theory (RTSDT) to obtain valuable insights into behavior of the structures. The study emphasizes the significant impact of magnetic field intensity and the ratio of PBR to NR on the natural frequency and loss factor of hybrid MR elastomer sandwich composite beams. Additionally, an exploration of factors such as ply orientation, boundary conditions, and elastomer core thickness allows for a comprehensive understanding of the variables affecting the dynamic properties of the sandwich composite structure. The study observes that natural frequencies increase with magnetic field intensity up to 600 MT, but decrease beyond that due to filler chain breakdown and particle agglomeration. Furthermore, an increase in PBR content notably improves damping properties, as evidenced by a significant increase in the loss factor. Similarly, an increase in NR content enhances stiffness, as seen in the increase in the fundamental natural frequency. Ply orientation is found to impact natural frequencies, with a significant 11% increase observed for the [0°/90°/0°]_s_ orientation. Finally, an increase in elastomer core thickness contributes to higher natural frequencies and improved damping. This research provides practical guidance for engineering applications, enabling the optimization of hybrid MR elastomer sandwich composite beams for various structural purposes. By tailoring factors such as magnetic field intensity, composition, and structural configurations, these materials can effectively enhance the performance, safety, and longevity of various structural systems and equipment.

## Figures and Tables

**Figure 1 polymers-15-04583-f001:**
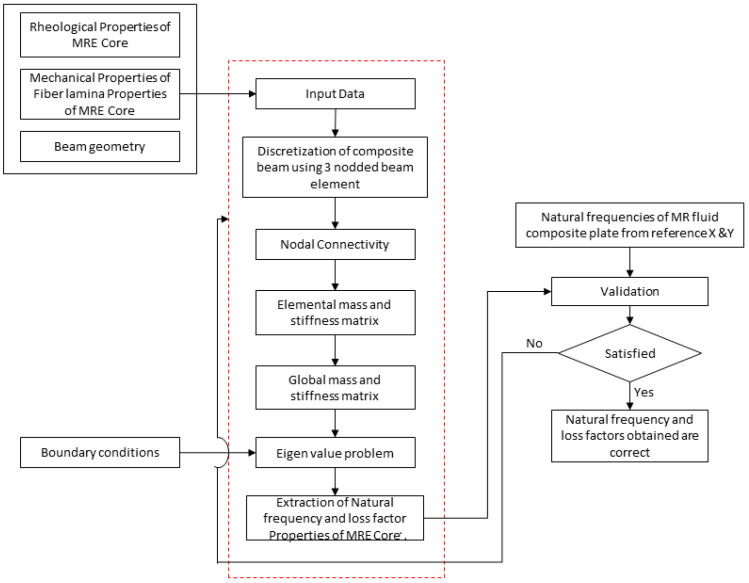
Steps involved in numerical investigation.

**Figure 2 polymers-15-04583-f002:**
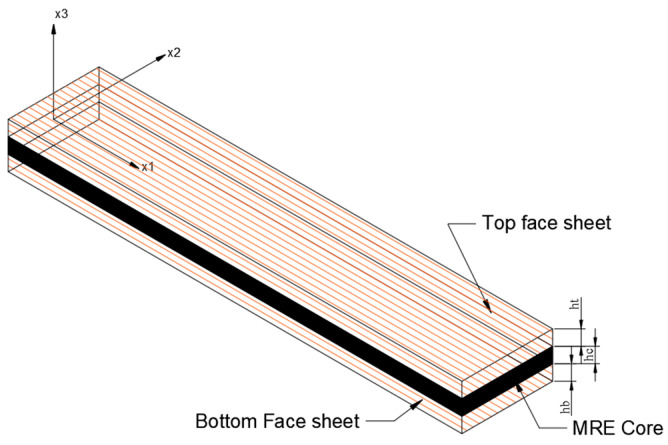
Dimensions of sandwich composite beam.

**Figure 3 polymers-15-04583-f003:**
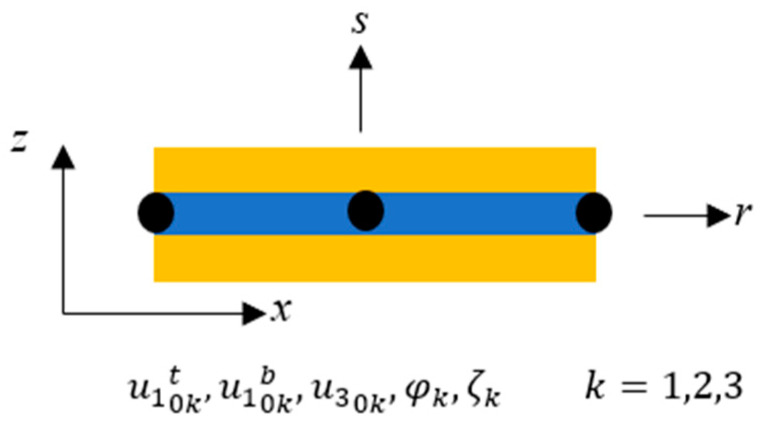
Three-noded beam element.

**Figure 4 polymers-15-04583-f004:**
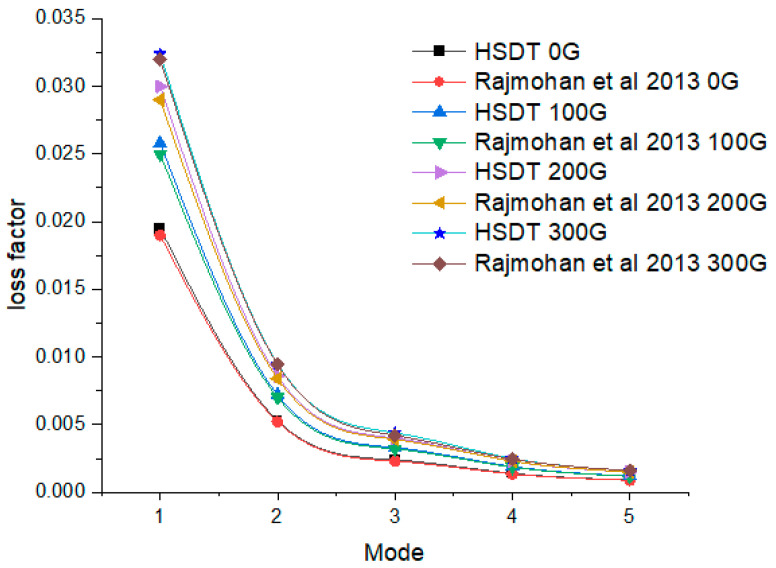
Variation in the loss factor of five modes under various magnetic fields [30].

**Figure 5 polymers-15-04583-f005:**
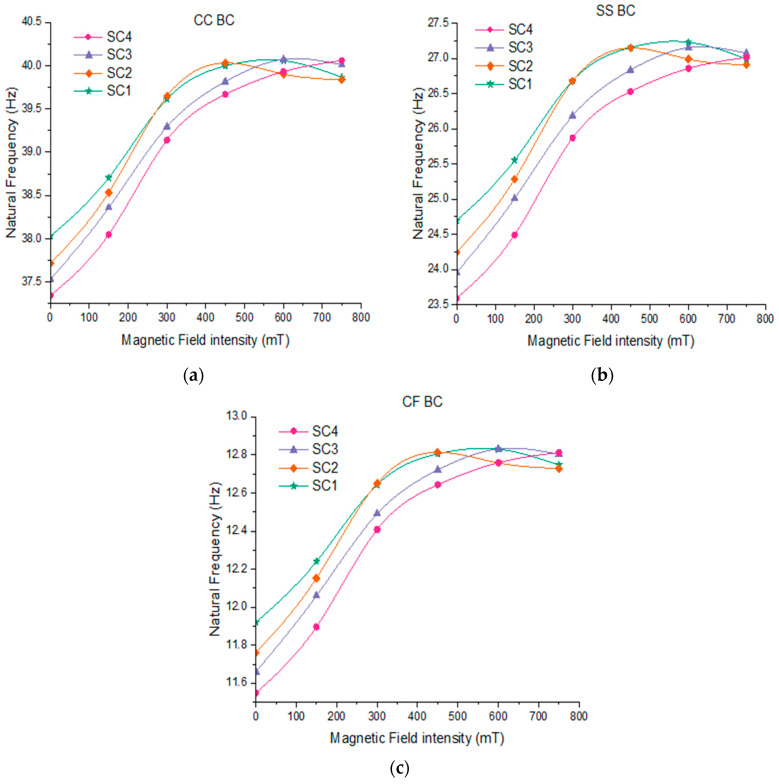
Influence of intensity of magnetic field on natural frequencies of sandwich beam under: (**a**) CC boundary condition; (**b**) SS boundary condition; (**c**) CF boundary condition.

**Figure 6 polymers-15-04583-f006:**
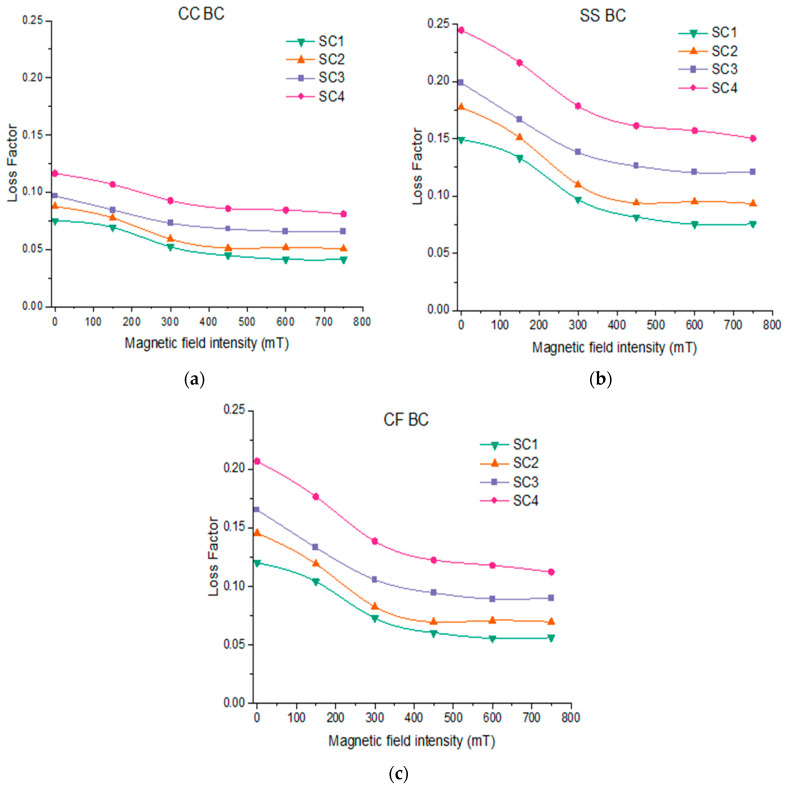
Influence of intensity of magnetic field on loss factor of sandwich beam under: (**a**) CC boundary condition; (**b**) SS boundary condition; (**c**) CF boundary condition.

**Figure 7 polymers-15-04583-f007:**
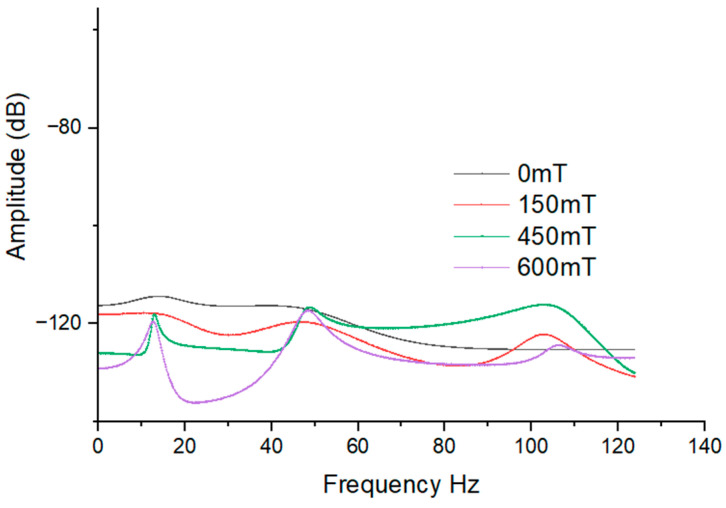
Transverse vibration response of hybrid MRE sandwich composite beam (SC1) under CF boundary conditions.

**Figure 8 polymers-15-04583-f008:**
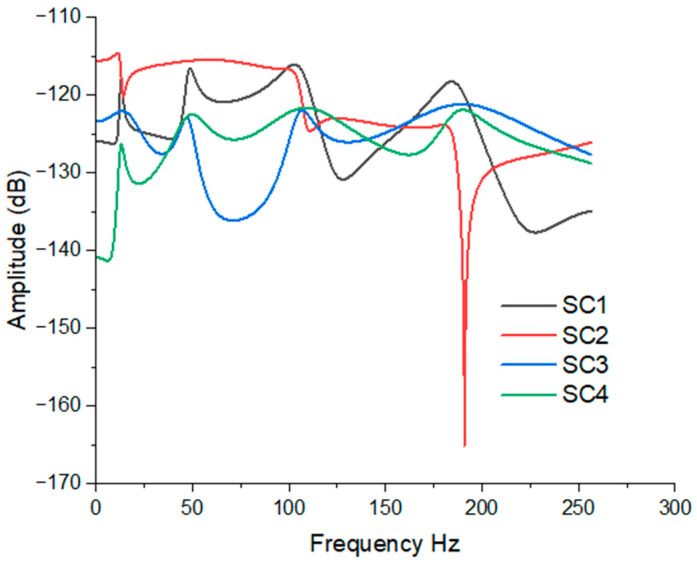
FRF plot of all configurations of hybrid MRE sandwich composite beam.

**Table 1 polymers-15-04583-t001:** Composition of MR elastomer core samples.

Materials	SC1	SC2	SC3	SC4
PBR	10	30	50	100
NR	90	70	50	0
Carbonyl Iron (CIP)	190	190	190	190
Cumarone	12	12	12	12
ZnO	5	5	5	5
4010NA	2	2	2	2
RD	3	3	3	3
Sulphur	3	3	3	3
SA	1	1	1	1
CZ	0.5	0.5	0.5	0.5

PBR: polybutadiene rubber; NR: natural rubber; CZ: n-cyclohexyl-2 benzothiazole sulfonamide; RD: poly(1, 2-dihydro-2, 2, 4-trimethyl-quinoline); ZnO: zinc oxide; SA: stearic acid.

**Table 2 polymers-15-04583-t002:** Comparison of natural frequencies of sandwich beam, determined from present FEM and that reported in [29,30].

Magnetic Field in Gauss	Modes	Natural Frequencies (Hz)				
		Present FEM	Ref. [29]	Ref. [30]	Error % [29]	Error % [30]
	1	38.84	40.34	40.74	3.72	4.66
	2	105.45	103.10	105.70	2.28	0.24
0	3	210.93	200.07	206.51	5.43	2.14
	4	356.45	332.45	344.72	7.22	3.40
	5	542.60	501.67	521.57	8.16	4.03
	1	51.57	50.92	51.88	1.28	0.60
	2	125.2	120.32	123.56	4.06	1.33
250	3	235.47	222.24	229.01	5.95	2.82
	4	383.33	357.33	369.67	7.28	3.70
	5	570.92	528.15	547.94	8.10	4.20

**Table 3 polymers-15-04583-t003:** Rheological properties of hybrid MRE samples under various magnetic fields.

Rheological Properties	Magnetic Field in mT	Hybrid MRE			
		SC1	SC2	SC3	SC4
G’ (MPa)	0	0.48	0.45	0.43	0.4
	150	0.53	0.51	0.49	0.45
	300	0.6	0.595	0.56	0.53
	450	0.63	0.625	0.6	0.57
	600	0.635	0.615	0.62	0.59
	750	0.62	0.61	0.615	0.6
ɳ	0	0.104	0.118	0.128	0.15
	150	0.1	0.11	0.118	0.144
	300	0.08	0.09	0.108	0.134
	450	0.07	0.08	0.104	0.128
	600	0.065	0.08	0.102	0.128
	750	0.064	0.078	0.102	0.124

**Table 4 polymers-15-04583-t004:** Properties of fiber lamina and density of the MRE core of various samples.

Lamina Properties	Density of MRE Samples
E_1_ = 30.5 GPa	
E_2_ = 6.99 Gpa	ρ_SC1_ = 2135.569 kg/m^3^
ν_12_ = 0.269	ρ_SC2_ = 2119.201 kg/m^3^
G_12_ = 2.8 Gpa	ρ_SC3_ = 2103.081 kg/m^3^
G_13_ = G_12_	ρ_SC4_ = 2063.836 kg/m^3^
G_23_ = 2.51 Gpa	
ρ = 1745 kg/m^3^	

**Table 5 polymers-15-04583-t005:** First five natural frequencies (Hz) of samples under various boundary conditions.

MRE Samples	BC	Modes	Magnetic Field Intensity					
			0 mT	150 mT	300 mT	450 mT	600 mT	750 mT
SC1	CF	1	11.92	12.24	12.64	12.81	12.83	12.75
		2	45.01	46.01	47.30	47.83	47.92	47.65
		3	101.10	102.63	104.69	105.55	105.69	105.25
		4	181.26	182.92	185.20	186.16	186.32	185.83
		5	287.63	289.37	291.77	292.79	292.95	292.44
	CC	1	38.03	38.70	39.61	39.99	40.06	39.86
		2	94.72	95.76	97.19	97.79	97.89	97.59
		3	176.34	177.59	179.31	180.05	180.17	179.80
		4	283.51	284.87	286.75	287.55	287.69	287.28
		5	416.34	417.77	419.76	420.61	420.75	420.32
	SS	1	24.70	25.55	26.68	27.15	27.23	26.99
		2	67.64	68.89	70.58	71.29	71.41	71.05
		3	135.56	137.01	139.01	139.86	140.00	139.57
		4	229.28	230.82	232.96	233.87	234.02	233.57
		5	348.98	350.58	352.79	353.74	353.89	353.42
SC2	CF	1	11.76	12.15	12.65	12.81	12.76	12.73
		2	44.53	45.74	47.34	47.87	47.69	47.59
		3	100.43	102.28	104.80	105.66	105.37	105.22
		4	180.69	182.70	185.48	186.45	186.12	185.96
		5	287.27	289.37	292.30	293.32	292.98	292.80
	CC	1	37.72	38.53	39.65	40.03	39.90	39.84
		2	94.32	95.58	97.32	97.93	97.72	97.62
		3	176.01	177.52	179.62	180.35	180.11	179.98
		4	283.36	285.00	287.30	288.11	287.84	287.70
		5	416.47	418.20	420.62	421.47	421.19	421.04
	SS	1	24.24	25.28	26.67	27.14	26.99	26.90
		2	67.05	68.56	70.64	71.35	71.11	70.99
		3	135.00	136.76	139.20	140.05	139.77	139.63
		4	228.89	230.76	233.37	234.28	233.97	233.82
		5	348.85	350.77	353.48	354.42	354.11	353.95
SC3	CF	1	11.66	12.06	12.49	12.72	12.83	12.80
		2	44.23	45.47	46.82	47.56	47.92	47.83
		3	100.05	101.92	104.03	105.21	105.79	105.65
		4	180.45	182.47	184.79	186.09	186.74	186.58
		5	287.25	289.36	291.79	293.16	293.85	293.68
	CC	1	37.53	38.36	39.29	39.81	40.07	40.01
		2	94.12	95.39	96.84	97.66	98.07	97.97
		3	175.92	177.43	179.18	180.17	180.66	180.54
		4	283.48	285.13	287.03	288.12	288.66	288.52
		5	416.88	418.61	420.62	421.76	422.33	422.18
	SS	1	23.96	25.01	26.18	26.83	27.15	27.07
		2	66.70	68.23	69.97	70.94	71.42	71.30
		3	134.73	136.50	138.54	139.68	140.25	140.11
		4	228.81	230.68	232.85	234.08	234.69	234.53
		5	349.03	350.96	353.20	354.47	355.11	354.95
SC4	CF	1	11.55	11.90	12.41	12.64	12.76	12.81
		2	43.91	44.97	46.57	47.33	47.70	47.88
		3	99.71	101.31	103.77	104.97	105.56	105.85
		4	180.49	182.20	184.89	186.21	186.87	187.19
		5	287.86	289.64	292.45	293.84	294.53	294.88
	CC	1	37.35	38.05	39.14	39.67	39.93	40.06
		2	94.03	95.11	96.80	97.63	98.04	98.24
		3	176.18	177.46	179.48	180.48	180.98	181.22
		4	284.30	285.69	287.88	288.98	289.52	289.79
		5	418.43	419.88	422.19	423.35	423.92	424.21
	SS	1	23.59	24.49	25.87	26.53	26.86	27.01
		2	66.33	67.64	69.66	70.64	71.14	71.37
		3	134.63	136.13	138.49	139.65	140.22	140.51
		4	229.19	230.77	233.28	234.52	235.14	235.44
		5	350.09	351.71	354.29	355.57	356.21	356.53

**Table 6 polymers-15-04583-t006:** Variation of fundamental natural frequency (Hz) of MRE samples with core thickness.

MRE Samples	Core Thickness in mm	Magnetic Field Intensity					
		0 mT	150 mT	300 mT	450 mT	600 mT	750 mT
SC1	1.5	10.6914	10.8794	11.10873	11.19967	11.21329	11.16621
	3	11.91927	12.23997	12.64273	12.80502	12.82992	12.74658
	4.5	12.44153	12.83503	13.33595	13.53934	13.5709	13.46684
SC2	1.5	10.60031	10.83163	11.11646	11.20739	11.17595	11.15912
	3	11.75898	12.15231	12.65093	12.81342	12.75771	12.72833
	4.5	12.2431	12.72488	13.34401	13.5478	13.4782	13.44174
SC3	1.5	10.5446	10.78049	11.02885	11.15951	11.22172	11.20608
	3	11.65949	12.05975	12.48997	12.72023	12.83099	12.80317
	4.5	12.11953	12.6092	13.14075	13.42756	13.56618	13.53139
SC4	1.5	10.48757	10.69432	10.99049	11.12393	11.1899	11.21911
	3	11.54746	11.89634	12.40732	12.6425	12.75894	12.81197
	4.5	11.97644	12.40191	13.0317	13.32452	13.46963	13.53654

**Table 7 polymers-15-04583-t007:** Variation in the loss factor of MRE samples with core thickness.

MRE Samples	Core Thickness in mm	Magnetic Field Intensity					
		0 mT	150 mT	300 mT	450 mT	600 mT	750 mT
SC1	1.5	0.080276	0.068653	0.047367	0.039047	0.035921	0.036456
	3	0.12041	0.10418	0.07294	0.0605	0.05571	0.05635
	4.5	0.140402	0.122066	0.085988	0.071504	0.065862	0.066524
SC2	1.5	0.097727	0.078866	0.053746	0.045009	0.045928	0.045254
	3	0.145748	0.119227	0.082726	0.069694	0.070963	0.069839
	4.5	0.169567	0.139482	0.097502	0.082351	0.083772	0.0824
SC3	1.5	0.111208	0.088485	0.069208	0.061286	0.057731	0.058316
	3	0.165264	0.133245	0.105829	0.094524	0.089416	0.090226
	4.5	0.192022	0.15564	0.1244	0.111516	0.105678	0.106586
SC4	1.5	0.139705	0.117939	0.091056	0.079836	0.076545	0.072718
	3	0.206857	0.176515	0.138642	0.122572	0.118034	0.112333
	4.5	0.240098	0.205741	0.162717	0.144349	0.139262	0.132634

**Table 8 polymers-15-04583-t008:** Effect of ply orientation on the fundamental natural frequency of the sandwich composite beam.

Magnetic Field Intensity mT	[0°/90°/0°]_s_				[90°/0°/90°]_s_				[0°/90°/45°]_s_			
	SC1	SC2	SC3	SC4	SC1	SC2	SC3	SC4	SC1	SC2	SC3	SC4
0	11.919	11.758	11.659	11.547	10.640	10.501	10.415	10.321	11.766	11.582	11.467	11.335
150	12.239	12.152	12.059	11.896	10.919	10.844	10.765	10.627	12.134	12.035	11.925	11.734
300	12.642	12.650	12.489	12.407	11.265	11.274	11.138	11.071	12.598	12.607	12.420	12.322
450	12.805	12.813	12.720	12.642	11.404	11.412	11.336	11.274	12.786	12.795	12.686	12.594
600	12.829	12.757	12.830	12.758	11.425	11.365	11.430	11.374	12.815	12.730	12.814	12.728
750	12.746	12.728	12.803	12.811	11.353	11.339	11.406	11.419	12.718	12.696	12.781	12.790

**Table 9 polymers-15-04583-t009:** Effect of ply orientation on the loss factor of the sandwich composite beam.

Magnetic Field Intensity mT	[0°/90°/0°]_s_				[90°/0°/90°]_s_				[0°/90°/45°]_s_			
	SC1	SC2	SC3	SC4	SC1	SC2	SC3	SC4	SC1	SC2	SC3	SC4
0	0.1204	0.1457	0.1652	0.2068	0.1184	0.1439	0.1637	0.2057	0.1392	0.1685	0.1912	0.2396
150	0.1041	0.1192	0.1332	0.1765	0.1016	0.1166	0.1307	0.1741	0.1204	0.1378	0.1541	0.2043
300	0.0729	0.0827	0.1058	0.1386	0.0703	0.0798	0.1027	0.1350	0.0843	0.0956	0.1224	0.1604
450	0.0605	0.0696	0.0945	0.1225	0.0581	0.0670	0.0911	0.1186	0.0699	0.0806	0.1093	0.1418
600	0.0557	0.0709	0.0894	0.1180	0.0534	0.0683	0.0859	0.1139	0.0644	0.0820	0.1034	0.1366
750	0.0563	0.0698	0.0902	0.1123	0.0542	0.0673	0.0868	0.1082	0.0651	0.0807	0.1043	0.1300

## Data Availability

The data used to support the findings of this study are included within the article.

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
