# Peer review of "Dynamic Behavior Modeling of Natural-Rubber/Polybutadiene-Rubber-Based Hybrid Magnetorheological Elastomer Sandwich Composite Structures"

_polymers, 2023, doi:10.3390/polym15234583_

Round 1

Reviewer 1 Report

Comments and Suggestions for Authors

The manuscript deals with an interesting research topic, but there is still room for further organizational, structural and argumentation improvements prior it to be accepted for publication at the Polymers and, to this end, the following review comments can be considered.

1. In the Methodology part of the analysis the following missing information can be added in form of two separated headings/sections referring to the fabrication of the magnetorheological elastomer material, as well as, fabrication of the MRE-sandwich structure.

2. A real photo/image of the preparing the sandwich composite structures can be provided in the form of a Figure.

3. Authors could provide, if  applicable, the transient response of the various vibration modes with the variation of magnetic field intensities should be added in: Time (s) - Amplitude graphs.

4. I am not fully convinced that the long arrangement of the mathematical equations could meet a comprehensive understanding from a wider readership, while it is also questionable what is the purpose of such an extensive calculation employment to a topic examined. I would recommend either a narrative shortening, or a much more extended narrative explanation on them, including all variables’ definition, values taken and units measured, for reproduction easiness from other researchers interested in it.

5. The Figures numbering need check and correction since Figures 2 and 5 are missing, while Figure 4 has been numbered twice, as:

Figure 4. Variation of loss factor of five modes under various magnetic field.

and later below

Figure 4. Influence of intensity of magnetic field on natural frequencies of sandwich beam under (a) CC boundary condition; (b) SS boundary condition; (c) CF boundary condition

6. In all cases where Figures contain subgraphs a, b, c, ….., then each one subgraph has to be explained in separate paragraph per subgraph (where applicable in the text), no all together in an aggregated manner.

7. In case of relevancy of elastic modulus and storage modulus with loss factor in a visualized depiction, this paired behaviour could be presented in the form of Figure(s).

8. All decimal numbers precision in the second half of Table 2, in Tables 4 and 5, has to be kept firmly and strictly to 2 digits, no more precision is needed here. Besides a more detailed and in depth explanation of the long arrangement of data presenting in Table 5 is highly recommended, making a more precise and straightforward understanding of the phenomena behind them.

9. At the end of the Discussion section authors can revisit and succinctly represent the fabrication constraints, the technological synergies/symbiosis prospected, the environmental concerns of materials handling/managing, the energy demands involved, to be all addressed. 2-3 sentences per each one of the aforesaid parameters, in a whole subsection of one extra text page, is adequate.

Reviewer 2 Report

Comments and Suggestions for Authors

The paper "Dynamic Behavior Modeling of Natural Rubber/Polybutadiene Rubber-Based Hybrid Magnetorheological Elastomer Sandwich Composite Structures" investigates the dynamic characteristics of a hybrid magnetorheological elastomer (MRE) sandwich composite structure. The study focusses on the dynamic behaviour of hybrid MRE sandwich composites. The aim of the paper is to understand the dynamic properties and behaviour of these materials that may be applied in various engineering fields.

However, the paper could be improved by addressing its shortcomings. For example, the paper does not clearly specify assumptions related to the studied theories, phenomenon and methodology. Furthermore, the paper does not provide a clear explanation of the assumptions, so the reader can guess on what basis these assumptions are based and possibly affect the interpretation of the results. The paper could also benefit from a more detailed discussion of the limitations of the study. Addressing these remarks could improve the clarity and reliability of the study.

Reviewer 3 Report

Comments and Suggestions for Authors

The article is devoted to dynamic properties of layered structure based on magnetorheological elastomer. The work can be divided into two parts. The first presented the analytical methodology and simulation method. The second part presented the results. The applied method of Reddy third-order shear deformation theory showed the quality releasable results about natural frequency of layered structure with different boundary conditions. The value of natural frequency and loss factors also depends on magnetic field, polymer composition and its shear modulus, thickness of MRE layer and orientation of magnetic field. The reviewer have two questions:

1)    The explanation and description of the Fig. 7 is: “In Figure 7, it is evident that there is a noticeable leftward shift in the natural frequency as the magnetic field intensity increases.” But one of the next sentence is: “As the magnetic field intensity increases, the natural frequencies shift to higher values”. Fig 7 and Fig 8 should be more clear for readers. For example, it should be pointed the values of natural frequencies or the value of magnetic field for Fig. 8.

2)    The formulas from (1) to (16) content a lot of variables and letter symbols without any descriptions and transcripts. The part with mathematical modelling is very complicated to read and understand. How the readers could test the model without explanation of each symbol?

The article can be published after revision.

Reviewer 4 Report

Comments and Suggestions for Authors

As for the viscoelastic effects on the deformation at high frequency

At high frequency region, the rubber material show viscoelastic behaviors and the rubber shows high loss modulus. The authors did not consider the effects of the frequency dependence of loss modulus or viscosity parts in the equation. They should consider the effects.

Reviewer 5 Report

Comments and Suggestions for Authors

I suggest to specify better the behavior of the increase of natural frequencies with magnetic field intensity up to 600 mT related to the filler.  

Round 2

Reviewer 1 Report

Comments and Suggestions for Authors

At this revised manuscript authors proceeded in a satisfactory and meticulous revision of their initial study, having all the research structure of their study well organized and its research parts fully developed. In this respect the revised manuscript sustains novel features and it can be accepted for publication at the Polymers journal as is.

Reviewer 2 Report

Comments and Suggestions for Authors

I would recommend to accept the paper in the present form.